# EFFICIENT CONTINUAL PRE-TRAINING FOR BUILDING DOMAIN SPECIFIC LARGE LANGUAGE MODELS

## ABSTRACT

Large language models (LLMs) have demonstrated remarkable open-domain capabilities. LLMs tailored for a domain are typically trained entirely on domain corpus to excel at handling domain-specific tasks. In this work, we explore an alternative strategy of continual pre-training as a means to develop domain-specific LLMs over an existing open-domain LLM. We introduce *FinPythia-6.9B*, developed through domain-adaptive continual pre-training on the financial domain. Continual pretrained FinPythia showcases consistent improvements on financial tasks over the original foundational model. We further explore simple but effective data selection strategies for continual pre-training. Our data selection strategies outperforms vanilla continual pre-training's performance with just 10% of corpus size and cost, without any degradation on open-domain standard tasks. Our work proposes an alternative solution to building domain-specific LLMs in a cost-effective manner.

## 1 INTRODUCTION

Large Language Models (LLMs) have exhibited a profound understanding of natural language, improving performance on an array of tasks (Brown et al., 2020). Using open web data has helped in creating general-purpose LLMs with a broad range of capabilities. General-purpose LLMs are however not "specialists"; for example, while LLMs could write good news articles, it would be hard-pressed to write specialized legal documents.

In order to make a specialist or domain-specific LLM, they need to be trained on domain data. Approaches for building domain-specific LLMs can be categorized into two categories: training domain-specific LLMs from scratch or using continual pre-training existing LLMs with domain data. Most researchers have taken the first approach of building domain-specific LLMs from scratch. Prominent examples are the Med-PaLM family (Singhal et al., 2022; 2023) for the medical domain, Galactica for scientific papers (Taylor et al., 2022), and BloombergGPT (Wu et al., 2023b) for finance. Little attention has been paid to building domain-specific LLMs using domain-adaptive continual pre-training, despite being a much cheaper alternative. Notably, PMC-LLaMA (Wu et al., 2023a), a medical LLM was trained through continual pre-training of LLaMA (Touvron et al., 2023) on medical papers. Continual pre-training can also be used for updating a LLM with the latest knowledge in an evolving environment.

In this work, we explore the following: 1) Is domain-adaptive continual pre-training helpful in building domain-specific LLMs?; 2) Can we employ data selection strategies for a more effective domain-adaptive continual pre-training?; and 3) Does domain-adaptive continual pre-training hurt LLM's open-domain capabilities? We answer these questions in the confines of finance domain by training a continually pre-trained model, FinPythia, built on top of Pythia (Biderman et al., 2023).

We report a boost on financial benchmarks (Xie et al., 2023) after continual pre-training on domain data of size 8% of what Pythia was trained on as an answer to the first question. We also observe an evidence of latest financial domain knowledge acquisition in FinPythia during qualitative analysis. To answer the second question, we propose two simple data selection techniques, task-aware *Efficient Task-Similar Domain-Adaptive Continual Pre-training* (ETS-DACP) and *Efficient Task-Agnostic Domain-Adaptive Continual Pre-training* (ETA-DACP). These methods outperform the performance of domain-adaptive continual pre-training with just 10% of selected domain data or 0.8% of Pythia's training corpus. We use three metrics for data selection: similarity, perplexity, and token type entropy. While similarity needs task data as seed data, the latter two metrics are task-agnostic metrics. To answer the third question, we benchmark these continually pre-trained models on four open-domain standard tasks like MMLU and TruthfulQA. We observe no significant performance change, indicating that LLM retains its general capabilities while adapting to the domain.

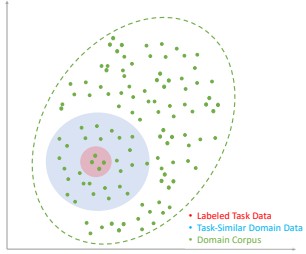

Figure 1: Relation between labeled task data, task-similar domain data and domain corpus in a manifold space.

The main contributions of this paper are threefold. Firstly, we curate a large-scale financial corpus comprising 16 billion words sourced from financial datasets. Secondly, our experiments demonstrate the promise of building domain-specific LLMs through continual pre-training, further validating and extending the findings obtained from smaller language models (Gururangan et al., 2020). This finding provides insights for building domain-specific LLMs with lower costs, as an alternative to expensive pre-training from scratch. Our results indicate that continual pre-training maintains the same open-domain performance as the original foundation model. Lastly, we propose two Efficient Domain-adaptive Continual Pre-training methods as a more efficient approach to vanilla continual pre-training. Our novel approach deploys data selection strategies that can achieve better performance with a fraction of the cost of the domain-adaptive continual pre-training.

## 2 METHODOLOGY

In this section, we describe the curation of our financial corpus used for continual pre-training, our domain-adaptive continual pre-training, task-adaptive continual pre-training, and our proposed task-aware domain-adaptive continual pre-training.

### 2.1 FINANCIAL CORPUS CURATION

In our evaluation of data sources, we consider three dimensions: public availability, licensing, and scale. We use two sources of data for the financial corpus: the financial news common crawl and SEC filings. Financial News CommonCrawl is curated by filtering out financial news from the public CommonCrawl data. We follow the de-duplication procedure of Pythia suite (Biderman et al., 2023) to remove duplicate training data. While there is conflicting evidence of duplication hurting the performance (Biderman et al., 2023; Lee et al., 2022), there is no evidence of the benefits of duplication in the training data. Hence, for a more efficient training, we use de-duplication following (Biderman et al., 2023). Using these two sources, we create a combined dataset of 23.9 billion tokens (16.5 billion words). Details of curation steps can be found in Appendix F.

### 2.2 DOMAIN-ADAPTIVE CONTINUAL PRE-TRAINING (DACP)

Typically, domain-specific LLMs are built by training the model from scratch using massive amounts of domain data. This procedure has two drawbacks: it is quite costly and needs much higher amounts of domain data, which is not as feasible in lower data domains like finance with very specialized and confidential data. Domain-adaptive continual pre-training (DACP) is a straightforward alternative to building from scratch; we continually pre-train a general-purpose LLM on a large scale corpus of domain-specific unlabeled data. Domain-adaptive continual pre-training has shown the ability to adapt the language models to better fit the in-domain distribution (Gururangan et al., 2020; Jin et al., 2022; Wu et al., 2022; Ruder & Plank, 2017). They also enable large language models to acquire new knowledge as new data appears (Jang et al., 2022b;a), instead of training the model from scratch. We use DACP in our experiments to benchmark its benefits.

### 2.3 TASK-ADAPTIVE CONTINUAL PRE-TRAINING (TACP)

Task-adaptive continual pre-training (TACP) refers to continual pre-training aiming to enhance performance on a targeted task. TACP has been studied in the context of smaller language models like BERT by pre-training the language model on labeled and unlabeled data from the task (Gururangan et al., 2020; Aharoni & Goldberg, 2020; Dai et al., 2019) showing improvements over the task. TACP has been used with pre-training loss objectives like Masked Language Modeling (MLM) loss on the training data of a downstream task for adapt the smaller language model to the downstream task

**without** using any task labels. While task data is usually quite limited, TACP has shown considerable effects on smaller language models like BERT. We benchmark TACP on our four financial evaluation tasks by continually pre-training the LLM on the task's data without any corresponding labels with a vanilla auto-regressive pre-training loss . Note, this is completely distinct than in-context learning or fine-tuning as no task labels are used in TACP; task training data is used as an unlabeled dataset.

## 2.4 TOWARDS AN EFFICIENT DOMAIN-ADAPTIVE CONTINUAL PRE-TRAINING

The primary limitation of TACP lies in its focus on constructing task-specific LLMs instead of foundation LLMs, owing to the sole use of unlabeled task data for training. While DACP uses a much larger domain corpus, it is prohibitively expensive. To address these limitations, we propose two approaches: *Efficient Task-Similar Domain-Adaptive Continual Pre-training* (ETS-DACP) and *Efficient Task-Agnostic Domain-Adaptive Continual Pre-training* (ETA-DACP). While ETS-DACP aims to build foundation LLMs for a set of tasks by tailoring the DACP to emphasize the significance of these tasks, ETA-DACP is more general and selects the most informative samples from the domain corpus for continual pre-training.

**Formulation**   We first formalize the problem. We are given an unlabeled domain pre-training corpus, $\mathcal{U}$ represented by green region in Figure 1. Next, we can take two scenarios: absence or presence of an unlabeled task corpus. The first scenario of the presence of a task corpus, which can be a single or group of tasks, $\mathcal{T}$ is depicted as the red region in Figure 1. Typically, the task corpus is a subset of the domain corpus, $\mathcal{T} \subset \mathcal{U}$, with $|\mathcal{U}| >> |\mathcal{T}|$. The goal of data selection is to select a subset, $\mathcal{D} \subset \mathcal{U}$, that is most helpful for pre-training the LLM model. We also assume that the selected domain corpus subset is much larger than the task corpus, $|\mathcal{D}| >> |\mathcal{T}|$, as is a typical case. The data selection problem can be formally defined as selection of optimal $\mathcal{D}^* \subset U$:

$$\mathcal{D}^* = \underset{\mathcal{D}^* \subset \mathcal{U}}{\operatorname{argmin}} \, \mathbb{E}_{x \in \mathcal{T}}[\mathcal{L}_t(y|f(\theta^*; x))] \tag{1}$$

where, $f(\theta; \cdot)$ is a LLM with parameters $\theta$, $y$ is the task output, $x$ is an input in target task data $\mathcal{T}$, and $\mathcal{L}_t$ is the target task loss or metric. $\theta^*$ is computed on pre-training task with $\mathcal{L}_{\text{pre-train}}$ as the pre-training loss, and $x_u$ as the unlabeled sample in $\mathcal{D}$:

$$\theta^* = \underset{\theta}{\operatorname{argmin}} \, \mathbb{E}_{x_u \in \mathcal{D}}[\mathcal{L}_{\text{pre-train}}(f(\theta; x_u))] \tag{2}$$

Our domain-adaptive continual pre-training can be viewed from the lens of unsupervised domain adaptation (Ganin et al., 2016). Our source data is the large unsupervised domain corpus, while the target data is the target task data. With pre-training, we do not have control over the alignment with task training data itself; our idea is that by aligning with the domain during pre-training, we could align the LLM with the task. This intuition is backed by evidence of LLM pre-training helping the performance over open domain tasks. We use the generalization bound from Ganin et al. (2016); Ben-David et al. (2010) since our problem is similar to unsupervised domain adaptation. Consider a hypothesis space $\mathcal{H}_p$ with $f \in \mathcal{H}_p$; generalization errors on source $\mathcal{D}$ and task data  as $\epsilon_{\mathcal{D}}$ and $\epsilon_{\mathcal{T}}$, respectively. The generalization bound can be given:

$$\epsilon_{\mathcal{T}}(f) \leq \epsilon_{\mathcal{D}}(f) + \frac{1}{2} d_{\mathcal{H}_p \Delta \mathcal{H}_p}(\mathcal{D}, \mathcal{T}) + \mathcal{C} \tag{3}$$

where, $d_{\mathcal{H}_p \Delta \mathcal{H}_p}$ is distribution discrepancy distance between $\mathcal{D}$ and $\mathcal{T}$ bounded by (Ganin et al., 2016):

$$d_{\mathcal{H}_p \Delta \mathcal{H}_p}(\mathcal{D}, \mathcal{T}) = \sup_{f, f' \in \mathcal{H}_p} |\mathbb{E}_{x \in \mathcal{D}}[f(x) \neq f'(x)] - \mathbb{E}_{x \in \mathcal{T}}[f(x) \neq f'(x)]| \leq 2 \sup_{\alpha(h) \in \mathcal{H}_d} [\alpha(h) - 1] \tag{4}$$

where, $\alpha(h)$ is optimal domain classifier and $\mathcal{H}_d$ is the hypothesis space of domain classifier. Zhao et al. (2017) prove that optimal state of minimum discrepancy distance $d_{\mathcal{H}_p \Delta \mathcal{H}_p}(\mathcal{D}, \mathcal{T})$ is when the domain classifier has random predictions achieving a state of highest entropy. We argue that it is achieved when the representations for samples in two domains are most similar, leading to a random domain classifier that is unable to distinguish between the two dataset distributions. Motivated by this intuition, we can use a strategy based on selecting samples with the most similar representations to our task dataset $\mathcal{T}$. We use the embedding similarity as a proxy for dataset similarity as getting the optimal representation is challenging in unpractical in the case of large corpus.

| Method | Data Selection Metric | Non-task Unlabeled Data | Task Unlabeled Data | Task Labeled Data |
|---|---|---|---|---|
| DACP | None | ✅ | ❌ | ❌ |
| TACP | None | ❌ | ✅ | ❌ |
| ETS-DACP | Similarity | ✅ | ✅ | ❌ |
| ETA-DACP-ppl | Perplexity | ✅ | ❌ | ❌ |
| ETA-DACP-ent | Entropy | ✅ | ❌ | ❌ |
| ETS-DACP-com | Similarity, Perplexity, Entropy | ✅ | ✅ | ❌ |
| (Instruction) Fine-Tuning | None | ❌ | ❌ | ✅ |

Table 1: Data selection metrics and data sources used for each method. ✅ indicates "yes", while ❌ indicates "no". Task unlabeled data refers to task data stripped off labels.

### 2.4.1 EFFICIENT TASK-SIMILAR DOMAIN-ADAPTIVE CONTINUAL PRE-TRAINING

We stipulate that we can form an optimal set $\mathcal{D}^*$ by selecting a portion of the domain data that is much closer to the task data (red) given by the blue region based on intuition before. We refer to this as *Efficient Task-Similar Domain-adaptive Continual Pre-training* (ETS-DACP). Fine-tuning LLMs can take a good amount of instructions, which are quite costly to create. ETS-DACP directly addresses this situation by using the relatively limited unlabeled task data to sample similar samples from the larger pool of pre-training domain corpus. We are motivated by prior research showing that unsupervised training on tokens that closely align with the target domain and tasks can lead to improved performance (Gururangan et al., 2020; Aharoni & Goldberg, 2020; Dai et al., 2019). Therefore, we hypothesize that continual pre-training LLMs on the unlabeled task data can be beneficial for target task performance as it adapts the model to the distribution of task tokens.

We use similarity between embeddings of task data and domain corpus samples to perform data selection. This allows us to select a subset from the domain corpus that closely resembles the distribution of task data. To quantify document-level task similarity, we employ cosine similarity between the document embedding and task data embedding using the *Spacy* model (Honnibal & Montani, 2017). This approach allows us to cost-effectively measure the alignment between task-specific information and the financial corpus, enabling more focused and targeted pre-training.

### 2.4.2 EFFICIENT TASK-AGNOSTIC DOMAIN-ADAPTIVE CONTINUAL PRE-TRAINING

While the previous case dealt with scenarios where task data is provided to us, in this method we explore scenarios where we do not have task data. This method also overcomes the limitation of ETS-DACP which makes the LLM too tuned to the task data instead of broader domain. We stipulate that two dimensions are important for obtaining domain information from a subset of pre-training domain data: **novelty** and **diversity**.

**Novelty** refers to the information that was unseen by the LLM before. We gauge the level of novelty in a document based on the **perplexity** recorded by LLM. Documents with higher perplexity are less represented in the original training corpus, thus being more likely to contain novel knowledge for the model. Such samples are also viewed as more difficult to learn (Bengio et al., 2009). Hence, these samples can be valuable in continual pre-training to help models acquire novel information.

Evaluating perplexity directly on the benchmark model incurs significant costs, as the inference requires approximately 25% of the training compute. To minimize this cost, we employ Pythia-70m as a surrogate model for computing document perplexity. Our preliminary experiment using a sample dataset reveals a strong correlation of 0.97 between the perplexity obtained from Pythia-1B and Pythia-70m. This high correlation justifies the use of a smaller model as a reliable surrogate, enabling more cost-effective sampling based on perplexity.

**Diversity** captures the diversity of distributions of token types in the domain corpus. Diversity has been shown to be an effective feature in related research on curriculum learning in language modeling (Tsvetkov et al., 2016; Ruder & Plank, 2017). We use part-of-speech (POS) tagging to get token types. Since entropy has been shown to be one of the best measures of diversity (Bengio et al., 2009), we use **entropy** of POS tags (Tsvetkov et al., 2016) as our diversity measure.

### 2.4.3 DATA SAMPLING STRATEGY

We proposed ETS-DACP and ETA-DACP to enhance vanilla DACP by refining the pre-training data through active selection of relevant samples. We can select the data in two ways:

**Hard Sampling:** We rank the samples in the domain corpus by the measure of choice. We select top-k samples from the domain corpus based on the metric(s), where $k$ is the number of samples needed to hit the pre-decided token budget for continual pre-training.

**Soft Sampling:** In this case, instead of giving binary weights by leaving out all the other examples in the corpus, we assign soft weights based on the distance metric. Taking similarity metric as an example distance metric, let's say a sample with a similarity score of 0.9. This similarity score would be normalized with the similarity scores for all other samples in the corpus. This normalized score is treated as the probability of selecting the sample as a part of the pre-training set. Once a sample is picked form the corpus, we remove the score for that sample from the sum of distance metric scores and recompute the probability for the remaining samples. This procedure allows for the continual pre-training to see the samples outside the blue region in Figure 1 as well, owing to non-zero probability of each sample, adding some diversity to the pre-training data.

We use the following three dimensions for selecting samples: similarity to task data (ETS-DACP), perplexity as a proxy for novelty (ETA-DACP), and diversity measured by token type entropy (ETA-DACP). In order to convert metric values into sampling probabilities, we propose a method based on quantile ranges. To achieve this, we first calculate the 0-100 quantiles for each metric within the training data. By dividing the range into 100 intervals using the 100 quantile values, documents are then assigned probabilities corresponding to the interval they fall into. This approach effectively normalizes our metrics, allowing for the aggregation of different metric types.

## 3 EXPERIMENTAL SETUP

### 3.1 EVALUATION TASKS

**Finance Domain Tasks** We evaluate the models on financial tasks to evalaute the effectiveness of our domain-adaptive continual pre-training. We adopt the *FLARE* framework (Xie et al., 2023) to evaluate our models. FLARE extends the LLM evaluation framework *lm-evaluation-harness*[1] by including various financial tasks. We follow their instruction prompt, data split, and metric computation for comparison. We consider following 4 tasks used in Wu et al. (2023b); Xie et al. (2023): (1) **Financial Phrase Bank**. FPB is a sentiment classification task on financial news (Malo et al., 2014). The sentiment reflects whether the news is considered as positive/neutral/negative by investors. (2) **FiQA SA.** An aspect based sentiment classification task based on financial news and headlines (Maia et al., 2018). (3) **Headline.** Binary classification task on whether a headline on a financial entity contains certain information (Sinha & Khandait, 2020). Each news article is associated with 9 tags like "price or not", "price up", "price down", "price stable", "past price", and "asset". (4) **NER.** Financial named entity extraction task is based on credit risk assessment section of SEC reports. Words in this task are annotated with PER, LOC, ORG, and MISC.

**General Domain Tasks** To evaluate the effect of domain training on the non-domain abilities, we test the performance on the following open domain tasks: (1) **ARC**: Abstraction and Reasoning Corpus Boratko et al. (2018) measures the ability of a model on predicting a correct output grid after a demostration of a task for the first time; (2) **MMLU**: Multi-task Language Understanding Hendrycks et al. (2020) tests knowledge of 57 tasks including elementary mathematics, history, and law; (3) **TruthfulQA**: Measures question answering ability of the model on 817 questions spanning 38 categories, some of which can be answered incorrectly based on human misconception found in common texts Lin et al. (2021). (4) **HellaSwag**: This benchmark Zellers et al. (2019) measures the commonsense ability of a LLM to generate a relevant follow up sentence, given an event description.

### 3.2 TRAINING SETUP AND INFRASTRUCTURE

For our benchmark pre-trained LLM model, we select 1B and 6.9B parameter models from the Pythia suite (Biderman et al., 2023). The Pythia model suite offers a diverse array of model sizes, ranging from 70 million to 12 billion parameters.

The continual pre-training configuration is tailored from Pythia's training setup reported in Biderman et al. (2023). Specifically, we set a learning rate of 1.2e-05 for FinPythia-6.9B and 3e-05 for *FinPythia-1B*, the smallest learning rates in their original schedules. We use small learning rates to mitigate catastrophic forgetting. We keep them constant throughout the course for efficient pre-training. We use the precision of bf16 rather than fp16 used in Pythia. We half the original batch size to 512.

We run the continual pre-training job on one P4d.24xlarge instance through AWS SageMaker. As the model size is moderate, we only use data parallelism via DeepSpeed ZeRO Stage 2 (Rasley et al., 2020) with activation checkpointing enabled. It takes 18 days for FinPythia-6.9B to pre-train and 3 days for FinPythia-1B to pre-train on 24 billion tokens.

---

[1]https://github.com/EleutherAI/lm-evaluation-harness

|  |  | BloombergGPT | OPT 7B | BLOOM 7B | GPT-J-6B | Pythia 1B | FinPythia 1B | Pythia 7B | FinPythia 7B |
|---|---|---|---|---|---|---|---|---|---|
| FPB | Acc | - | 57.22 | 52.68 | 50.21 | 42.85 | 47.14 | 54.64 | **59.90** |
|  | F1 | 51.07* | **65.77** | 52.11 | 49.31 | 43.94 | 46.52 | 55.79 | 64.43 |
| FiQA SA | Acc | - | 40.43 | **70.21** | 60.42 | 54.51 | 46.13 | 60.85 | 52.34 |
|  | F1 | 75.07* | 31.29 | **74.11** | 62.14 | 56.29 | 44.53 | 61.33 | 53.04 |
| Headline | F1 | 82.20* | **62.62** | 42.68 | 45.54 | 44.73 | 53.02 | 43.83 | 54.14 |
| NER | F1 | 60.82* | 41.91 | 18.97 | 35.87 | 49.15 | **55.51** | 41.60 | 48.42 |
| Average | F1 | 67.29* | 50.40 | 46.97 | 48.22 | 48.53 | 49.90 | 50.64 | 54.83 |
| $\Delta(\%)$ | Avg-F1 | - | - | - | - | 0.00 | 2.82% | 0.00 | 8.27% |

Table 2: 5-shot results on financial tasks from domain adaptive continual pre-training. * indicates that the results are extracted from BloombergGPT (Wu et al., 2023b), which are evaluated with different prompts and data split. The values is not directly comparable to others. **Bold** indicates the best results among all the evaluated models except BloombergGPT. Underline indicates the better results between FinPythia and Pythia of the same sizes. $\Delta(\%)$ is the % difference between the Average F1 of four tasks with FinPythia and Pythia of same model sizes.

# 4 RESULTS AND ANALYSIS

## 4.1 DOMAIN-ADAPTIVE CONTINUAL PRE-TRAINING

To evaluate financial domain tasks, we compare FinPythia with Pythia and other open-sourced models of similar size. We include OPT-7B (Zhang et al., 2022), BLOOM-7B (Scao et al., 2022), and GPT-J-6B (Wang & Komatsuzaki, 2021) as benchmark models. While we report results from open-sourced models, the main insights are obtained from the comparison between Pythia and FinPythia, as their difference reflect the effect of domain-adaptive continual pre-training. Models are evaluated in a 5-shot setting for each task. Shots are randomly sampled from the tasks' training dataset for each test instance following FLARE (Xie et al., 2023) benchmark.

Results are reported in Table 2. FinPythia-6.9B and FinPythia-1B exhibit superior performance on tasks FPB, Headline, and NER while showing comparatively lower results on the FiQA SA task compared with Pythia counterparts. DACP boosts the average task performance by 2.8% for the 1B model and 8.3% for the 6.9B model. These outcomes directly substantiate the impact of domain-adaptive continual pre-training on enhancing in-domain task performance. Furthermore, Pythia-6.9B outperforms OPT-7B, BLOOM-7B, and GPT-J-6B on average.

*Comparison with BloombergGPT*: results reported on FLARE are not directly comparable with results reported in BloombergGPT (Wu et al., 2023b), as the data splits used by them are not public. We could not match the performance of publicly available models like OPT-66B or GPT-NeoX reported by (Wu et al., 2023b), on all four tasks. See the detailed comparison in Appendix A.

## 4.2 EFFICIENT DOMAIN-ADAPTIVE CONTINUAL PRE-TRAINING

FLARE uses 5-shot in-context performance over the entire training data, *i.e.,* each test sample while evaluating each model sees different train samples. This also makes it harder to compare different models, as each test example sees completely different 5 training examples across models during inference. Practically, we have limited set of labeled samples — there is no luxury of seeing entire large training data. We observed a high standard deviation owing to this randomness in selection across the large training dataset. *To overcome this randomness and make the comparisons fair across models, we set aside a pool of 50 labeled data samples from the training dataset for each task, referred to as the "shot pool".* For the remaining training samples, we remove their labels and utilize them as unlabeled task data, which is used in our data selection strategy utilizing task data. This particular configuration is adopted because we do not have access to unlabeled task data to evaluate the efficacy of TACP. By using this setup, we also simulate the constraints posed by scarce labeled data. Although this approach creates unlabeled task data for TACP, the size is too small, containing only 0.24 million tokens from the four tasks.

Using Efficient DACP methods, we select 10% subset of the financial corpus for each method. We also create another version of ETS-DACP called **ETS-DACP-com** by using the other two measures with similarity by averaging all three measures for ranking/weighting. To mitigate overfitting, both the TACP and Efficient DACP methods run for a single epoch, employing the same pre-training configuration as DACP to ensure a fair comparison. We run these experiments with Pythia-1B due to the compute budget. We perform the evaluation ten times using different random seeds and report the mean performance for each of our four financial tasks.

The evaluation results are presented in Table 3. While TACP shows significant improvement in model performance compared to the original Pythia-1B, ETS-DACP stands out as the top-performing approach among DACP, TACP, and efficient DACP methods in terms of average task performance. This enhanced performance cannot be solely attributed to the increased number of tokens, as DACP with the same amount of tokens yields inferior results. The results underscore the efficacy of both

| | Tokens | FPB F1 | FiQA SA F1 | Headline F1 | NER F1 | Average F1 | Win Rate (%) | Δ(%) |
|---|---|---|---|---|---|---|---|---|
| **Pythia 1B** | 0 | 52.84 (15.5) | 65.32 (13.7) | 45.61 (10.0) | 48.77 (13.7) | 53.14 (7.5) | 45.5 | 0.00 |
| **DACP** | 2.39B (10%) | 64.77 (10.4) | 59.85 (19.0) | 41.41 (6.5) | 51.32 (7.6) | 54.34 (8.9) | 59.1 | 2.25% |
| **DACP (FinPythia 1B)** | 23.9B (100%) | 59.16 (12.1) | 52.84 (18.1) | 53.34 (9.4) | **55.20** (5.8) | 55.14 (2.5) | 52.3 | 3.76% |
| **TACP** | 0.24M | 66.80 (10.5) | 72.27 (2.2) | 38.91 (1.5) | 50.55 (11.7) | 57.13 (13.2) | 56.8 | 7.50% |
| **Hard Sampling** | | | | | | | | |
| **ETS-DACP** | 2.39B (10%) | 67.11 (9.6) | 50.84 (21.9) | **71.56** (7.1) | 49.52 (8.4) | **59.76** (9.7) | **63.6** | **12.45%** |
| **ETA-DACP-ppl** | 2.39B (10%) | **73.66** (1.9) | 45.86 (24.9) | 39.11 (2.0) | 48.69 (8.5) | 51.83 (13.1) | 40.9 | -2.4% |
| **ETA-DACP-ent** | 2.39B (10%) | 69.58 (8.4) | 58.14 (19.1) | 59.83 (11.1) | 46.18 (15.7) | 58.43 (8.3) | 61.4 | 9.99% |
| **ETS-DACP-com** | 2.39B (10%) | 62.58 (14.7) | **72.83** (1.8) | 53.91 (11.6) | 48.34 (15.9) | 59.41 (9.3) | 61.4 | 11.79% |
| **Soft Sampling** | | | | | | | | |
| **ETS-DACP** | 2.39B (10%) | 72.45 (3.4) | 47.08 (18.1) | 40.82 (7.9) | 46.16 (15.1) | 51.63 (12.3) | 34.1 | -2.84% |
| **ETA-DACP-ppl** | 2.39B (10%) | 61.44 (18.4) | 52.44 (13.6) | 41.00 (5.6) | 43.80 (13.7) | 49.67 (8.0) | 20.5 | -6.5% |
| **ETA-DACP-ent** | 2.39B (10%) | 68.20 (9.5) | 57.00 (22.5) | 62.06 (11.4) | 38.00 (19.6) | 56.31 (11.3) | 52.3 | 5.96% |
| **ETS-DACP-com** | 2.29B (10%) | 64.41 (11.0) | 67.97 (9.2) | 51.22 (12.5) | 47.68 (13.8) | 57.82 (8.6) | 52.3 | 8.80% |

Table 3: Effect of TACP and efficient DACP measured by 5-shot results on financial task for Pythia-1B class of models. The reported are mean and standard deviation (in parenthesis) of 10 runs. ETA-DACP-ppl is ETA-DACP with perplexity measure, and ETA-DACP-ent is with entropy measure. ETS-DACP-com is task similar DACP with data selection by averaging all three metrics: perplexity, similarity, and entropy. Win rate is percentage of times a model is more accurate than other models in a pair-wise comparison (Liang et al., 2022). **Bold** indicates the best results and underline indicates the second best per task. Δ(%) denotes the % difference between a CL method versus base Pythia 1B's average F1. Note, the results in third row (DACP 100%) correspond to FinPythia 1B results in Table 2. The difference in results between the two tables is because we fixed the pool of 5-shot examples to make a fair comparison between all the experiments versus the standard way of randomly selecting 5-shots in previous works Wu et al. (2023b).

task-adaptive and domain continual pre-training LLMs on unlabeled task data, in line with results observed in other model types (Aharoni & Goldberg, 2020; Gururangan et al., 2020).

We can observe the following: 1) ETS-DACP trained on 10% outperforms DACP with 100% of the data; 2) ETS-DACP has the best performance among all three counterparts and is on par with a combination of three metrics - ETS-DACP-com; 3) ETA-DACP-ent trained on 10% corpus is a close second despite not having any access to task data, handily surpassing DACP trained on 100% of the data; and 4) Efficient DACP methods with hard sampling outperform ones with soft sampling.

These results clearly show that ***not all data is equal*** *for continual pre-training*. In fact, all the data used in efficient DACP methods (10%) is a subset of the data in DACP. Since DACP's (100%) performance is lower than ETS-DACP/ETA-DACP-ent, adding more data on top of highly similar or high entropy data actually hurts the performance. The difference in results between hard and soft sampling adds more evidence to this observation. While there is variability across tasks, on an average, adding examples from outside the top decile of metrics hurts the performance with the notable exception of ETS-DACP-com which is a combination of all three metrics. Hence, we should carefully curate the data for any domain continual pre-training.

Note, 10% of domain data (2.39B) translates to less than 1% of the 300 billion tokens base Pythia was trained on. Hence, being selective during the data curation process for continual pre-training can have large effects on domain performance at a small cost. These results demonstrate the effectiveness of continual pre-training on domains and task (sub-domains). A natural question that arises from this exercise is ***whether the LLM is losing its generality by being further tuned on a narrow domain?*** In short, is the LLM becoming a specialist at the expense of being a generalist? We answer this question by measuring the performance of continually pre-trained LLM variants on out-of-domain tasks which Pythia was evaluated on. Table 4 shows the performance on the standard four non-finance tasks. We do not observe any significant change in the performance on the four out-of-domain tasks except for DACP with 100% data. Hence, *by being selective about the data to use for continual pre-training, we can keep the LLM's original capability intact while improving their domain performance.*

### 4.3    ABLATION ON PERCENTAGE OF PRE-TRAINING DATA SELECTED

We show ablation with percentage of pre-training data in Figure 2. We see ETS-DACP and ETA-DACP-ent methods saturate near an average F1 score of 59% at 5% of pre-trained data and start declining after using 10% of pre-trained data. This shows that adding samples that are not as informative, drops the performance as LLM learns over not so useful examples, adjusting its distribution. For DACP, we observe a constant increase in performance. We observe an interesting trend in perplexity with a higher performance than DACP with 1% of selected pre-trained data. It starts to drop significantly afterwards, reaching the lowest performance at 5% of the training data, and recovering afterwards. On further investigation by randomly sampling the perplexity based data

selected between 1%-5% of pre-training sample region, we saw a particularly examples with long tables, devoid of natural language text. This change in distribution versus the rest of pre-training corpus and tasks could explain the drop in performance with perplexity based data selection.

**Comparison of Data Selection Metrics** From the results in Table 1 and Figure 2, we can observe that task similarity based selection works the best, in line with intuition presented in Equation 4: similarity of training data with the task data is most beneficial while (pre-)training the LLM. Entropy is second best but most effective task-agnostic data selection technique for domain pre-training. Highest entropy samples are selected based on the named entity distribution; these samples would have a more variety of domain specific entities like names and locations, versus lower entropy samples. Our hypothesis is that such samples expose LLM to more domain knowledge than the lower entropy samples with lower type of different entities and hence, less informative. Perplexity exhibits an interesting phenomenon: there is benefit with an initial set of top-1% perplexity samples but not any further. High perplexity samples are more novel for the model but novelty can come both from out-of-distribution as well as lower quality samples. We do observe mostly high quality financial articles in top-1% of perplexity samples while in top 1% to top 5% perplexity range, we observed samples with long tables which arguably are noisy on what the foundational model has been trained on. Hence, perplexity has a high probability of being affected by the noise in the datasets which scores high on perplexity metric versus entropy metric.

Since, most of large datasets can have noisy samples, perplexity based data selection is not a good idea. Correlation between perplexity and other two metrics: similarity (0.21) and entropy (0.14) is quite low. Hence, these two avoid selecting these noisy samples. Given that we generally want our domain LLMs to perform well over unseen tasks, adapting the pre-training to a task agnostic framework is better. *Based on our experiments, entropy metric scores high both on task agnostic as well as performance on downstream tasks.*

## 5 RELATED WORK

**Domain specific large language models.** While the majority of released LLMs are general-purpose models, domain-specific LLMs have emerged as valuable counterparts. Google's MedPaLM and MedPaLM-2, trained on a medical domain corpus, achieved state-of-the-art results on medical benchmarks (Singhal et al., 2022; 2023). Bloomberg developed the first financial LLM from scratch by training on a financial corpus (Wu et al., 2023b) while Galactica was developed for scientific domains (Taylor et al., 2022). Continual pre-training presents an alternative approach to building domain-specific LLMs from scratch. Wu et al. (2023a) build medical LLMs through continual pre-training LLaMA (Touvron et al., 2023) on medical papers. However, they do not evaluate the model's quantitative performance in a non-fine tuning setting. In this work, we measure the model's performance in an in-context learning setting, showing the clear benefits of continual pre-training.

**Continual pre-training of Language Models.** Continual pre-training of language models on unlabeled data for a given task has been demonstrated to be beneficial for enhancing end-task performance (Aharoni & Goldberg, 2020; Gururangan et al., 2020; Ke et al., 2023). In scenarios involving domain shift, domain-adaptive pre-training bears similarities to task-adaptive pre-training to some extent. Aharoni & Goldberg (2020) have documented that continual pre-training a model on a similar domain contributes to improved task performance on the target domain. Notably, the work closest to ours is presented in Gururangan et al. (2020), which shows that continual pre-training

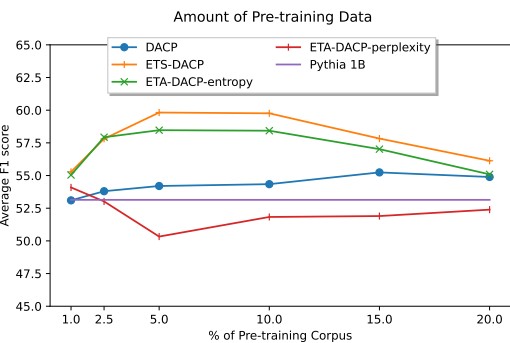

Figure 2: Ablation of amount of pre-training data versus Average F1 score for continual pre-training methods.

| | Tokens | ARC | | MMLU | | TruthfulQA | | HellaSwag | | Average | | $\Delta(\%)$ |
|---|---|---|---|---|---|---|---|---|---|---|---|---|
| | | Acc | Acc Norm | Acc | Acc Norm | MC1 | MC2 | Acc | Acc Norm | Acc | Acc Norm | |
| **Pythia 1B** | 0 | 25.94 | 29.27 | 26.29 | 26.29 | 23.62 | 40.47 | **37.65** | **47.83** | 28.38 | **35.96** | 0.00 |
| **DACP** | 2.39B (10%) | 26.28 | 29.44 | 26.43 | 26.43 | 24.48 | 42.26 | 36.83 | 45.34 | 28.50 | 35.87 | 0.42% |
| **DACP (FinPythia 1B)** | 23.9B (100%) | 24.32 | 27.47 | 26.09 | 26.09 | 24.60 | 42.05 | 35.34 | 42.45 | 27.59 | 34.52 | -2.78% |
| **TACP** | 0.24M | 25.34 | 28.41 | 24.93 | 24.93 | 24.48 | 41.95 | 37.03 | 47.27 | 27.95 | 35.64 | -1.51% |
| **Hard Sampling** | | | | | | | | | | | | |
| **ETS-DACP** | 2.39B (10%) | 24.74 | 28.07 | 25.99 | 25.99 | 23.26 | **43.85** | 36.31 | 44.79 | 27.57 | 35.68 | -2.85% |
| **ETA-DACP-ppl** | 2.39B (10%) | **26.71** | 28.41 | 26.31 | 26.31 | 24.97 | 41.42 | 36.70 | 44.89 | **28.67** | 35.26 | 1.02% |
| **ETA-DACP-ent** | 2.39B (10%) | 25.34 | 27.99 | 24.60 | 24.60 | 24.11 | 41.38 | 36.92 | 44.98 | 27.75 | 34.74 | -2.21% |
| **ETS-DACP-com** | 2.39B (10%) | 26.37 | 29.35 | 26.58 | 26.58 | 24.48 | 41.51 | 36.61 | 44.97 | 28.51 | 35.60 | 0.45% |
| **Soft Sampling** | | | | | | | | | | | | |
| **ETS-DACP** | 2.39B (10%) | 26.45 | 28.33 | **27.10** | **27.10** | 24.60 | 41.73 | 36.24 | 44.49 | 28.60 | 35.41 | 0.77% |
| **ETA-DACP-ppl** | 2.39B (10%) | 25.85 | **29.69** | 26.59 | 26.59 | 24.85 | 42.17 | 36.55 | 44.71 | 28.46 | 35.79 | 0.28% |
| **ETA-DACP-ent** | 2.39B (10%) | 25.94 | 29.10 | 25.61 | 25.61 | 24.60 | 41.64 | 36.78 | 45.20 | 28.23 | 35.39 | -0.52% |
| **ETS-DACP-com** | 2.39B (10%) | 25.77 | 27.47 | 27.05 | 27.05 | 24.24 | 41.82 | 36.93 | 44.62 | 28.50 | 35.24 | 0.42% |

Table 4: Evaluation on standard tasks **Bold** indicates the best value for a column. We follow the evaluation practice used to create HuggingFace Open LLM leaderboard. $\Delta(\%)$ is the percentage difference between average accuracy of Continually pre-trained LLM and Pythia 1B.

of language models on both unlabeled task data and augmented unlabeled task data, sampled from the in-domain corpus based on similarity. While these works use task data, we also propose a task agnostic method, ETA-DACP, as task similarity is prohibitively expensive for LLMs. DAS (Ke et al., 2023) built on top of DGA (Ke et al., 2022), a state-of-the-art continual pre-training method on language models, is too expensive for LLMs. These methods cost even more than vanilla pre-training over the entire pre-training corpus, as they use multiple forward-backward passes. DAS (Ke et al., 2023) calculates KL divergence over embeddings from language model (RoBERTa) with two dropout settings for a given unlabeled sample to calculate the importance of a sample and contrastive learning over the previously learnt knowledge, in addition to the forward-backward pass of conventional full pre-training. Hence, a total of three forward-backward passes over the entire pre-training corpus — 300% the cost of vanilla continual pre-training, which is impractical for LLMs given the costs. In contrast to all these approaches, we introduced simple methods that save the cost of pre-training — 10% of cost of vanilla continual pre-training — by pre-selecting the data before the actual pre-training instead of doing importance weighting during pre-training (Ke et al., 2023; 2022).

**Data selection.** Data selection for continual pre-training plays a critical role in choosing the most valuable samples for the training process. Various linguistic features independent of specific domains or tasks have been shown to be beneficial for data selection and learning curricula (Ruder & Plank, 2017; Tsvetkov et al., 2016). In the context of LLMs, there is a limited understanding of how to curate data for pre-training, let alone for continual pre-training. *To best of our knowledge, ours is the first work that attempts to do data selection in the context of LLMs for effective continual pre-training.*

## 6 CONCLUSION

In this paper, we demonstrate the efficacy of domain-adaptive continual pre-training for developing domain-specific LLMs. Our results in the finance domain show that domain-adaptive continual pre-training improves the LLMs' performance on financial tasks. Domain-adaptive continual pre-training enables the LLMs to acquire new knowledge in the financial domain at a much lower cost.

Furthermore, we propose efficient domain-adaptive continual pre-training methods, ETS-DACP and ETA-DACP, to enhance the effectiveness of the continual pre-training. By being selective during the training data curation, our methods refine the continual pre-training, yielding even better results with just 10% of the data and cost of vanilla continual pre-training. ETA-DACP with data selection based on task-agnostic measures like entropy works almost at par with the task-aware data selection strategy. This finding can be used to build data selection for continual pre-training even in the absence of task data. We also observe no degradation in performance on open-domain standard tasks, implying that domain-adaptive continual pre-training does not hurt the model's open-domain capabilities.

Our findings place domain continual pre-training as a strong alternative to building domain-specific LLMs from scratch. By being smarter about data selection for continual pre-training, we can surpass vanilla continual pre-training at a fraction of the cost. There is a wide belief in LLM community that just throwing more data during pre-training helps, our results suggest that quality of data matters too. While domain continual pre-training has been studied widely in small language model literature, we provide a unique insight in LLMs given the scale and costs involved. Overall, our work paves the way for developing domain-specific LLMs at a reduced cost, with implications for a wide range of LLM applications.

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

## A    BENCHMARK BLOOMBERGGPT'S PERFORMANCE

As BloombergGPT is evaluated using an in-house data split, and the calculation details of reported metrics may not be identical, direct comparisons of their results with ours are not feasible. To adequately assess the efficacy of continual pre-training, we benchmark BloombergGPT's performance against the FLARE framework. This involves evaluating OPT-66B and GPT-NeoX-20B's performance, as obtained from FLARE, and comparing it to the results reported in Wu et al. (2023b). This rigorous benchmarking ensures a fair and comprehensive evaluation, providing valuable insights into the effectiveness of our continual pre-training approach in relation to financial LLMs trained from scratch.

|  |  | FLARE | | BloombergGPT | |
|---|---|---|---|---|---|
|  |  | GPT-NeoX | OPT-66B | GPT-NeoX | OPT-66B |
| FPB | F1 | 46.75 | 40.00 | 44.64 | 48.67 |
| FiQA SA | F1 | 73.86 | 37.36 | 50.59 | 51.60 |
| Headline | F1 | 62.62 | 61.36 | 73.22 | 79.41 |
| NER | F1 | 47.03 | 52.24 | 60.98 | 57.49 |
| Average | F1 | 57.57 | 47.74 | 57.36 | 59.29 |

Table 5: Evaluation results obtained on FLARE benchmark versus BloombergGPT (Wu et al., 2023b) for two public models: GPT-NeoX and OPT-66B.

Table 5 reports the comparison results. GPT-NeoX reports similar average task performance under two evaluation frameworks, but its performance on individual tasks varies. For example, the F1 score on FiQA SA obtained by FLARE is 46% higher than BloombergGPT's evaluation, whereas F1 scores for Headline and NER are lower. Moreover, OPT-66B reports inferior results based on FLARE than BloombergGPT's evaluation on all of the 4 tasks, and the average task performance is 20% lower. These results suggest that BloombergGPT's evaluation results are inflated compared with FLARE. The comparison is still inconclusive unless BloombergGPT is benchmarked on FLARE or BloombergGPT's evaluation configuration is made public.

## B    QUALITATIVE EVALUATION

Qualitative examples generated by Pythia-6.9B and FinPythia-6.9B are presented in Table 6. Upon examination, we observe that FinPythia-6.9B exhibits a superior ability to generate more relevant and detailed responses for finance-related questions. For example, in first example, while Pythia exhibits no knowledge of very finance domain specific term "SPAC" and starts hallucinating, FinPythia gives a correct definition of SPAC. Same with the other three examples. These findings suggest that the continual pre-training helps FinPythia-6.9B acquire in-domain knowledge, lacking in Pythia.

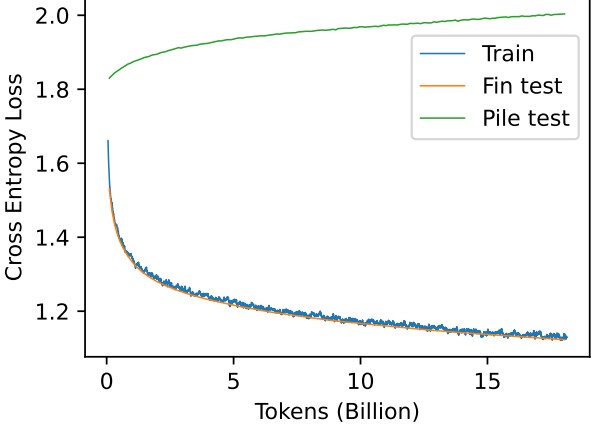

Figure 3: Training loss of FinPythia-6.9B. FinPythia-6.9B achieves significant loss drop in financial corpus at mild expense of Pile loss with vanilla pre-training (DACP).

---

**Question:** What is a SPAC?

**Pythia6.9B:** Spac is an acronym for Space. It is defined as either (1) above, just beneath, below, above, or through a port in the body of a planet, Moon, or other heavenly body; or (2) beyond a star's sphere of influence, usually in interplanetary space or interstellar space; also, (3) outside the limits of the Milky Way Galaxy.

**FinPythia-6.9B:** A SPAC is a type of merger and acquisition vehicle that is used to raise money from investors, use that money to identify a company that could be valuable to an interested party, then merge with the company and take it public. The use of the SPAC is to give investors a piece of the upside in the acquired company and allows the company to issue shares that may give investors a piece of the upside of the company as well.

**Question:** Who is Cathie Wood?

**Pythia-6.9B:** Cathie Wood is the founder of All Things Organic, an organic marketing company that she founded in 2000. In addition to being an organic marketer, Cathie is also a Certified Coach, foodie, cook, a past lover of yoga, a certified teacher, mother and wife to three gorgeous children.

**FinPythia-6.9B:** Cathie Wood is an American investor and advisor, the founder and chief executive officer of Ark Investment Management.

**Question:** How do you like Fed's latest policy to fight inflation?

**Pythia6.9B:** I like Fed's latest policy to fight inflation, which was reported by Tyler. Fed's latest policy was reported in a meeting on September 12 and it was named as SOMO, which stood for Standardized Open Market Operations. According to the policy, the U.S. central bank could buy $40 billion worth of financial instruments if the price of the instruments reach 6 percent.

**FinPythia-6.9B:** There is a big confusion about Fed's new policy to fight inflation. They are following a new policy from September 2022 called the fastest quarter-percentage point rate hikes in two decades to fight the "costly" inflation which is the 8.5 percent rate of increase in food and energy prices that may aggravate inflation and lead to recession in the near future.

**Question:** What is the difference between bond and stock in terms of risk profile?

**Pythia-6.9B:** Stock is less risky (you can liquidate your shares anytime) but bond is more risky (you can't liquidate).

**FinPythia-6.9B:** When you buy a bond, you receive a set of fixed payments from the issuer at a regular interval over a fixed term. You take on the full risk of the coupon payments not being made, so a bond is always considered to be a lower risk debt instrument than a stock. However, a stock is a more liquid investment than a bond. When buying a stock, you are buying shares in the company that the bond represents and you hold on to the stock until you want to cash out, and it can be used as collateral to borrow money.

---

Table 6: Generation examples from Pythia-6.9B and FinPythia-6.9B. We observe FinPythia's updated financial events knowledge post 2021, providing factual answers with reasoning as shown in the last example.

## C   TRAIN AND TEST LOSS OF CONTINUAL PRE-TRAINING METHODS

To monitor the pre-training process, we randomly sample 0.1% of our financial corpus as a financial test dataset. The model is also evaluated on the Pile test dataset. The loss trajectory for FinPythia-6.9B is reported in Figure 3. The training loss in the figure is smoothed using a moving average of 50 optimization steps. We observe a sharp decrease in Financial test (Fin test) loss during the early stage of continual pre-training, and the progress gradually becomes saturated, similar to the loss trajectory of training from scratch (Wu et al., 2023b; Touvron et al., 2023). The loss log suggests that domain-adaptive continual pre-training succeeds in adopting Pythia to the financial domains at the expense of a mild increase in Pile loss (Pile test).

We show the plots of Finance domain loss (Fin Test) and open domain loss (Pile Loss) for our efficient DACP methods in Figure 4. ETS-DACP-com (Hard sampling) has the lowest loss for Fin Test loss as it uses both task knowledge and also uses high entropy/perplexity samples in the the larger financial pile. This selection difference is illustrated in Figure 5. All methods have similar Fin Test loss for Soft sampling as we sample entire financial corpus space for sampling, allowing the model to see the entire space of corpus (green dots in Figure 6) mimicking Figure 5b . This effect of sampling is further seen in the Hard Sampling case; ETS-DACP limited to samples in the blue shaded region in Figure 5a, has a higher fin test loss as well as Pile test loss, as it is confined to the task distribution unlike task agnostic methods which see the wider financial corpus distribution as shown in Figure 5b. ETA-DACP-ent and ETA-DACP-ppl show similar loss curves as expected as they both sample from the entire finacial corpus. ETS-DACP-com has a higher loss than these but lower loss than ETS-DACP, as it is a mixture of these three sampling techniques.

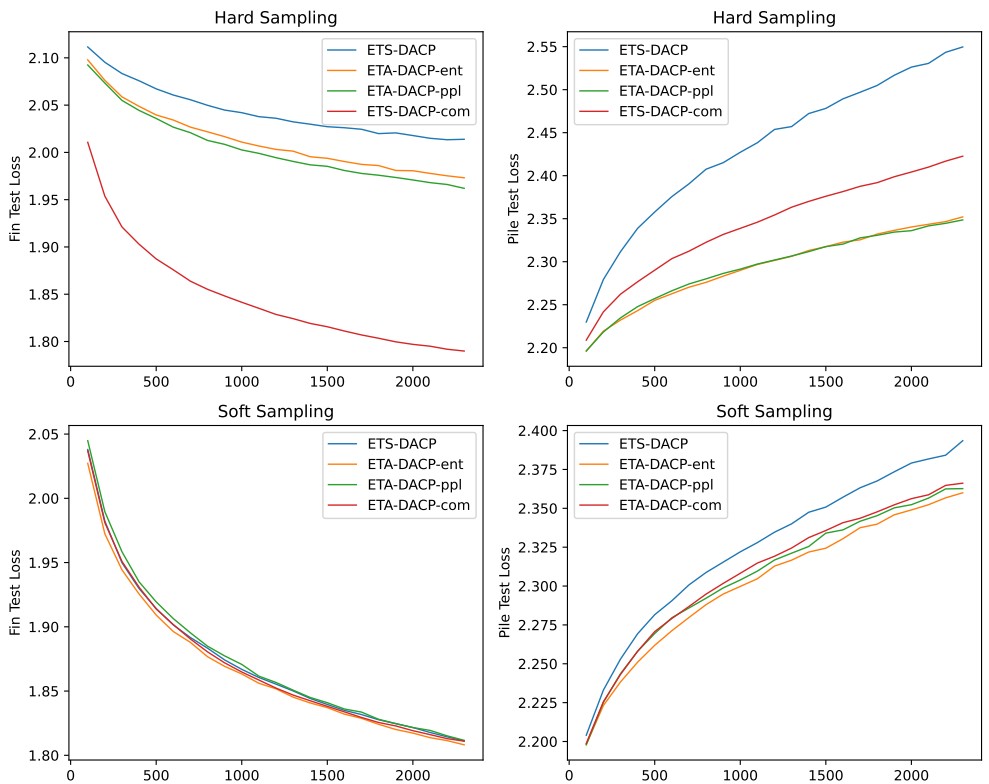

Figure 4: Loss curves for in domain loss (Fin Test loss) on left and general domain loss (Pile loss) on right for our Efficient DACP class of methods. X-axis is number of epochs ran for 10% of domain corpus.

ETS-DACP has the highest loss for open domain Pile loss. However, we did not observe any significant degradation of performance on open domain tasks with ETS-DACP. Surprisingly, there is a tight correlation between losses of ETS-DACP-ent and ETS-DACP-ppl, while ETS-DACP-ppl performs consistently and considerably worse than ETS-DACP-ent on our tasks. These observations suggest that there is no good correlation between actual our task performance and loss curves. Using validation/test loss with unlabeled data is not a good proxy for task performance, atleast in this domain. This is supported by Liu et al. (2023)'s observations on low correlation between task performance and pre-training loss.

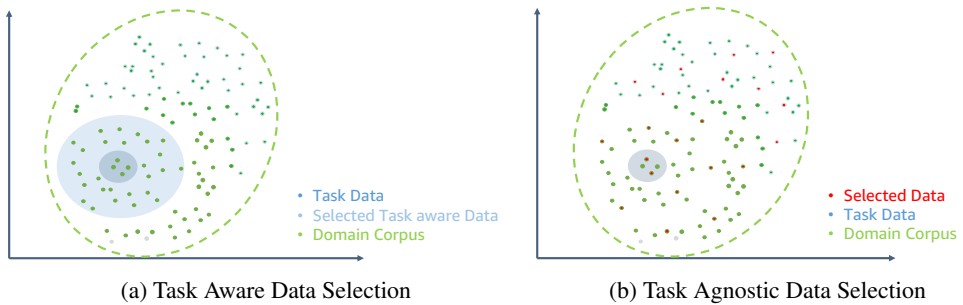

(a) Task Aware Data Selection        (b) Task Agnostic Data Selection

Figure 5: Pictorial depiction of Task Aware Data Selection (left) versus Task Aware Data Selection (Right). While task aware data selection confines the model to only see data in the regions similar to task data (or a sub-domain), task agnostic allows the model to see the wider distribution of domain data.

# D  PERPLEXITY, SIMILARITY, AND DIVERSITY

In this section, we present an in-depth analysis of the distribution of perplexity, similarity, and diversity within our financial corpus. Our findings reveal that all three metrics display a highly skewed distribution. Specifically, as illustrated in the top row of Figure 6, the similarity metric demonstrates a two-modal pattern, potentially attributable to the presence of two distinct sources within our financial corpus.

Figure 7 shows the Spearman's rank correlation of all three metrics. We see that the three metrics exhibit low correlation. This suggests that subsets of data we selected by ranking across these three metrics do not have a high degree of overlap. This inspired us to create the ETS-DACP-com method, which combines the three metrics together to balance the three different dimensions. Figure 8 shows the quantile distribution of three metrics for selected subsets for each of the efficient DACP methods with hard sampling.

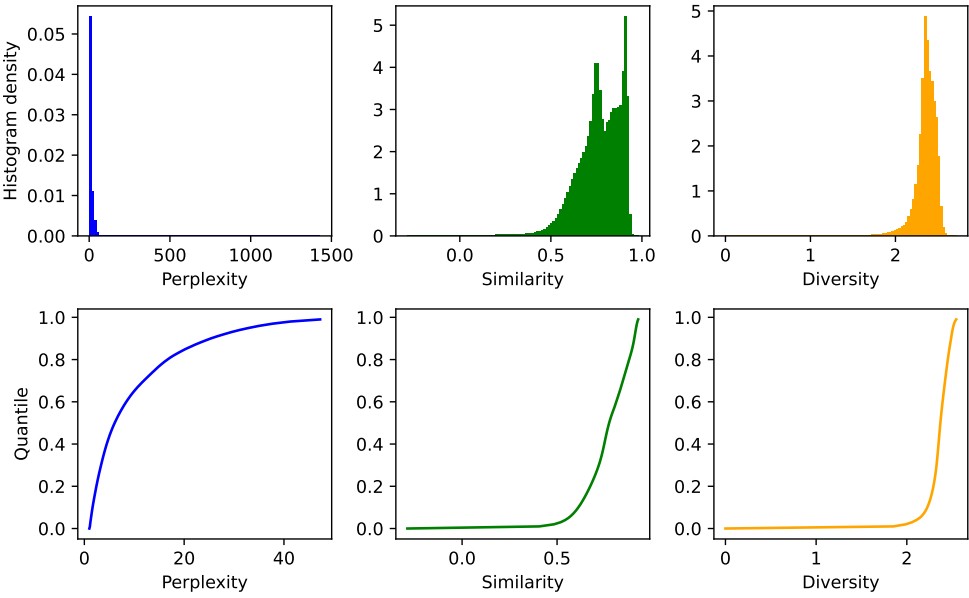

Figure 6: Distribution of perplexity, similarity and diversity.

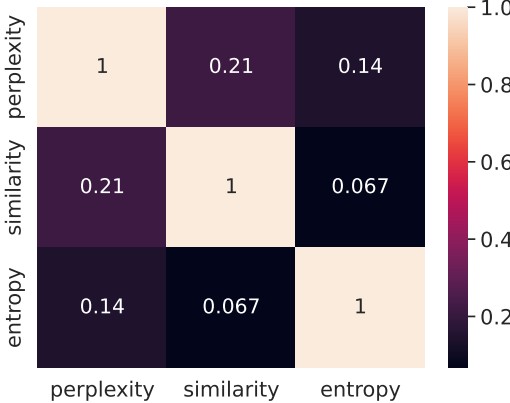

Figure 7: Spearman's rank correlation heatmap between perplexity, similarity, and entropy measures.

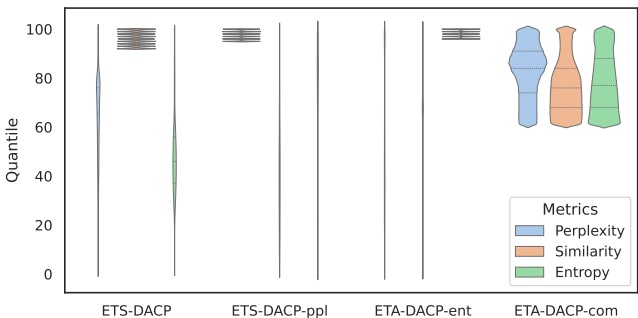

Figure 8: Average sample quantile of subsets of financial corpus used in ETS-DACP-com and ETS-DACP.

## E   ETS-DACP-COM VS ETS-DACP

ETS-DACP-com effectively strikes a balance between constructing a domain-specific LLM and a task-specific LLM. To demonstrate its efficacy, we utilize the average quantile of similarity, knowledge novelty, and diversity as the sampling weights. By applying these weights, we perform weighted sampling, selecting 10% and 20% of the financial corpus without replacement to construct the training data.

The average sample quantile for various subsets of the financial corpus is illustrated in Figure 8. We claim that using a simple average of quantiles for the three metrics achieves a good balance among the three dimensions—the average quantile for the three dimensions lies in a similar ballpark for each subset. In contrast, the subset for ETS-DACP exhibits higher perplexity and lower or middle entropy, suggesting that unlabeled task data contains new knowledge but is less diverse. For ETA-DACP-ppl and ETA-DACP-ent, the samples are uniform across the other two dimensions.

## F   FINANCIAL DATASET CURATION

We describe the two data sources for curating our domain corpus: Financial News CommonCrawl and SEC filings.

**Financial News CommonCrawl [13.2B words, 83.5%]**   We curate an English financial news dataset by pre-processing the publicly available News CommonCrawl dumps hosted on AWS S3[2] spanning from 2016 to 2022. To identify financial news articles from the vast collection of News CommonCrawl dumps, we employ two filtering mechanisms: the domain filter and the URL keyword filter. Firstly, we establish a comprehensive portfolio of web domains corresponding to reputable news outlets that predominantly focus on financial, economic, and business news, such as CNBC. We retain news articles specifically sourced from these financial news domains, which constitute a substantial portion of our financial corpus.

Secondly, to capture financial articles from general news outlets, we observe that many of them designate dedicated sections or subdomains for business, economy, or finance news, like Fox Business. To effectively identify these financial articles, we implement a simple yet effective keyword-based approach that targets financial sections and subdomains within general news outlets. The filtering processes ensure the selection of a financial corpus appropriate for our continual pre-training in the financial domain.

**SEC Filing [3.3B words, 16.5%]**   Public companies in the United States are legally required to submit their financial statements on a regular basis. The Securities and Exchange Commission (SEC) facilitates public access to these filings through the Electronic Data Gathering, Analysis, and Retrieval (EDGAR) System, which has been available since 1993. On average, this system accommodates approximately 40,000 new files per year. To enrich our financial corpus, we include 10-K filings from the period spanning 1993 to 2022. To ensure data accuracy and consistency, these filings are parsed and pre-processed using the package detailed in Loukas et al. (2021). Furthermore, we optimize

---
[2]s3://commoncrawl

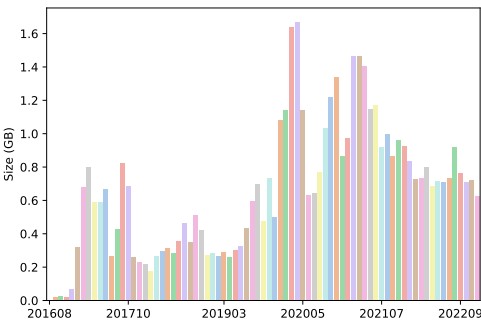

Figure 9: Financial news size by month

the quality of our corpus by eliminating report sections containing less than 20 words, to remove spurious examples.

**List of Domains used to Filter Financial News**   We use the following keywords to identify subdomains and urls: economy, market, finance, money, wealth, invest, business, industry.

