# OpenReview forum: "Efficient Continual Pre-training for Building Domain Specific Large Language Models"
_ICLR.cc/2024/Conference — ICLR 2024 Conference Withdrawn Submission_

### Official Review · Reviewer_rYGF · 2023-10-26

**Soundness:** 2 fair
**Presentation:** 2 fair
**Contribution:** 3 good
**Rating:** 5
**Confidence:** 4

**Summary:**

The paper investigates a new approach to developing domain-specific Large Language Models (LLMs) using continual pre-training. A domain-specific LLM, FinPythia-6.9B, was created for the financial sector through domain-adaptive continual pre-training. The results show that FinPythia has improved performance on financial tasks compared to the original base model. Additionally, the paper proposes effective data selection strategies at the embedding level for continual pre-training.

**Strengths:**

* The paper proves that continual pre-training can facilitate the LLM's performance on domain-specific LLMs.

* The authors use embedding level selection to acquire the essential data samples and show that with 10% data, the pre-training can achieve comparable performance instead of using a large amount of data.

**Weaknesses:**

* This paper only demonstrates that the proposed pipeline can be efficient with the data selection on one type of LLM, Pythia, which is insufficient to support the claim of efficiency advantages for other types of LLMs, e.g., LLAMA, OPT.

* The comparison with other baseline methods is not fair because more data is used for training in continual pre-training. To illustrate the effectiveness of the continual pre-training, the authors should apply the proposed method to other types of LLMs.

**Questions:**

* Could you do similar continuous pre-training and ablation studies for LLAMA-7B?

* Could you do the same continuous pre-training for OPT, BLOOM, and GPT-J and report the results on financial tasks?

---

> ### Author Response · Authors · 2023-11-18
> **Continually Pre-training other LLMs**
>
> We would like to thank the reviewer for their insightful comments and taking the time to review our work.
>
> We agree with the reviewer that our work only demonstrates the proposed methods on Pythia and not on any other LLMs, which is unfair to other LLMs like OPT. However, the main goal of our paper was not to compare between different LLMs but to demonstrate:
>
> * **Effectiveness of continual pre-training**:  The focus of the paper is the comparison between Pythia and FinPythia models, with FinPythia performing better than Pythia.
> * **Effectiveness of Data selection methods**: We demonstrate that using data selection methods, we can perform than the vanilla pre-training.
>
>
> **Response to questions**
> While we would love to do continual pre-training and ablation or LLaMA 7B and OPT/BLOOM, the costs for doing so are quite prohibitive. It took us nearly 20 days to train Pythia 7B model on one P4.24xlarge instance we have access to. We had to scale down to Pythia 1B for our data selection methods because of these costs, to compare between our proposed data selection methods. Note, GPT-J was trained by EleutherAI as well, the predecessor of Pythia. The main reason of selecting Pythia was its availability of different sized models: 14 m to 12B which allowed us this flexibility of comparing continual pre-training at different sizes as well as out results with perplexity selection, as perplexity from Pythia 70M model has a correlation of 0.97 with Pythia 1B/7B models, allowing us for easy data selection. This flexibility is not available in other model families, where 7B is typically the smallest model size.
>
> We hope this clarifies the reviewers concerns about not doing continual pre-training on other LLMs.

---

> > ### Comment · Reviewer_rYGF · 2023-11-22
> >
> > Thank you for providing further clarification. Nevertheless, the absence of ablation studies involving various other Large Language Models (LLMs) limits the strength of the argument for the generalizability of your proposed data selection method. Given this consideration, I have decided to lower my scores.

---

> ### Author Response · Authors · 2023-11-22
>
> We would like to request the reviewer to re-consider their revision. Absence of ablation study with other LLMs does not limit the strength of this work, as the main claim of the paper is the effectiveness of the data selection method results presented in Tables 3 and 4. **We are NOT making making a claim that other LLMs when continually pre-trained would do better or worse than FinPythia**. Table 2 results of comparison with other LLMs (BLOOM, OPT, GPT-J, BloombergGPT) is just presented for benchmarking; it does not add or take away from the central claims of this work.
>
> Doing a continual pre-training of LLaMA-2, BLOOM, GPT-J does not have a bearing on our claims. Our central claim is how performing data selection increases the effectiveness of domain continual pre-training. **So the main comparison is with all the various data selection methods by fixing a base model (Pythia)**.
>
> We would also like to emphasize about the costs involved here to run each data selection (10 rows in Table 3) for each model of size 7B. All the techniques we have presented (similarity-based, entropy-based, and perplexity-based data selection) use smaller models to be of practical use to typical researchers with low compute resources, democratizing LLMs. We're unable to perform the experiments with 7B models with our limited resources. **This raises the bar too high for any work in this space and  disadvantages researchers in LLM community not having access to significant compute resources at their disposal**. This is what this paper is helping to address by presenting a method that allows researchers to create domain LLMs without having to train from scratch which only industrial labs with million dollar budgets can do like BloombergGPT.
>
> We would really appreciate if the reviewer can re-consider their decision, as this work opens up the space for researchers with limited compute budgets.

---

> > ### Comment · Reviewer_rYGF · 2023-11-22
> >
> > Apologies for any confusion earlier. The rationale behind suggesting experiments with other types of LLMs, e.g., LLAMA, is to underscore that the proposed data selection methods are broadly effective across various prevalent models. I know it is hard to conduct experiments with limited resources. However, the generalizability of a data selection method is essential. It is necessary to demonstrate further the effectiveness of the data selection method for various LLMs. Additionally, to increase the contribution of this paper, the experiments for LLMs with larger sizes should be conducted as well when resource and time constraints are less of a concern.

---

> ### Author Response · Authors · 2023-11-22
>
> Thank you for your clarification. We agree that these experiments would strengthen the paper but we do think that it would be too strict to reject the paper because of these experiments. Most of the papers that have been accepted in the past in ICLR use a single LLM model for their experiments. For instance:
>
> - Chain-of-thought reasoning [1] only shows the CoT on GPT-3 with in-context learning
> - Compositional task representations [2] demonstrate the experiments on a T5 with in-context learning
> - Compositional Semantic parsing with LLMs [3] uses InstructGPT for all the experiments on a single task of semantic parsing
> - Ke et al [4] only used RoBERTa as the base model for continual pre-training
>
>
> Notably, none of these experiments in these papers needed high computational resources unlike our case for pre-training as they use in-context learning or smaller models.
>
> We think this increases the bar too high, limiting the exploration in this area by rejecting the work due to lack of these additional experiments. In an era, where large industrial research labs are not openly sharing details about pre-training, too high barriers would be demotivating for open source community with limited resources.
>
> We appreciate your time and responses, and would request you again to re-consider, if possible. Thank you once again for your thoughtful review and responses.
>
>
> [1] Zhang, Z., Zhang, A., Li, M. and Smola, A.. Automatic Chain of Thought Prompting in Large Language Models. ICLR 2023.
>
> [2] Shao, N., Cai, Z., Liao, C., Zheng, Y. and Yang, Z.. Compositional task representations for large language models. ICLR 2023.
>
> [3] Drozdov, A., Schärli, N., Akyürek, E., Scales, N., Song, X., Chen, X., Bousquet, O. and Zhou, D.. Compositional Semantic Parsing with Large Language Models. ICLR 2023.
>
> [4] Ke, et al., Continual Pre-training of Language Models. ICLR 2023.

---

### Official Review · Reviewer_npUz · 2023-10-29

**Soundness:** 2 fair
**Presentation:** 2 fair
**Contribution:** 2 fair
**Rating:** 5
**Confidence:** 4

**Summary:**

This paper explores domain-adaptive pre-training in the finance sector. It introduces a heuristic data selection method based on novelty (perplexity) and diversity (entropy of POS tagging). The results demonstrate its superior performance compared to general language models (LLM).

**Strengths:**

1. DAPT/DACP is an important and practical problem
2. The proposed data selection method is simple to use and easy to understand

**Weaknesses:**

In this paper, Domain-adaptive pre-training (DAPT) or DACP is not a novel concept, and the main innovation lies in the proposed data selection method. However, the paper lacks comparisons with other DAPT baselines, which is a significant drawback. For example, some prior works have explored modifications to the DAPT loss or gradient. Notably, [1] addresses continual pre-training but is not compared with in this work, nor are any other mentioned baseline systems in [1].

[1]: Adapting a Language Model While Preserving its General Knowledge, Ke et al., EMNLP 2022

**Questions:**

See above

---

> ### Author Response · Authors · 2023-11-18
> **Comparison with Ke et al**
>
> We really appreciate reviewer’s feedback on our paper and pointing out a significant weakness in the paper.
>
> We agree with the reviewer that Domain-adaptive pre-training (DAPT) is not a novel concept and has been studied in the realm of traditional language models. These works [1, 2], however are limited to using smaller LMs like Roberta for continually pre-training on much smaller pre-training datasets like ACL papers.
>  * These methods are not feasible for LLMs because of cost and run-time considerations, as they select sample importance while pre-training. For example,  DAS [1] costs 300% over the vanilla pre-training, while our method costs 10% of vanilla pre-training. Not to mention the difficulty in infrastructure needed: DAS/DGA needs two copies of the model in memory as they calculate the importance of a sample based on different dropout ratios for the same model to have comparable run times to conventional pre-training. Otherwise, the run time for pre-training is in the orders of vanilla pre-training’s run time by swapping different configurations of the model sequentially.
> * While our work does not have the complexity, the challenges from creating one of the largest domain specific corpus and the costs of comparing different methods of data selection at this scale in our resources were not trivial. Our work takes a novel approach of filtering the samples before the pre-training versus filtering while pre-training, which is not feasible at the scale of LLM pre-training. Our method costs less than 3.33% of DAS/DGA.
>
> We have added this discussion in Section 5 under continual pre-training of language models section. Thanks once again for pointing this out, and we hope this clarifies why our work while being simpler is significant for LLM field.
>
>
>
> [1] Ke, et al. "Continual Pre-training of Language Models." ICLR. 2023.
>
> [2] Ke, et al. "Adapting a Language Model While Preserving its General Knowledge." EMNLP. 2022.

---

### Official Review · Reviewer_LVHJ · 2023-10-30

**Soundness:** 3 good
**Presentation:** 3 good
**Contribution:** 3 good
**Rating:** 6
**Confidence:** 5

**Summary:**

This paper introduces FinPythia-6.9B, a LLM developed through domain-adaptive continual pre-training on the financial domain. The paper shows that continual pre-training can yield consistent improvements on financial tasks over the original model. It also experiments different data selection strategies for continual pre-training and proposes a data selection strategy based on novelty and diversity measurements. The proposed efficient domain-adaptive continual pre-training technique outperforms vanilla continual pre-training's performance with just10% of corpus size and cost.

**Strengths:**

1. The paper contributes FinPythia-6.9B a foundation model for financial domain via continual pre-training. FinPythia-6.9B outperforms the original LLM on a series of tasks in the financial domain which showcases the feasibility of building domain-specific LLMs in a cost-effective manner.
2. Improving continual pre-training from the data selection aspect is interesting. This paper conducts extensive experiments on different data selection methods and the gained insights can be useful to the community. Also, the proposed data selection technique is effective based on the experimental results.
3. The paper also curates a large-scale financial corpus.

**Weaknesses:**

1. The tasks for experimental are mainly classification tasks, which is limited as the LLM is powerful and should be evaluated on more complicated tasks or at least some generation tasks. I know the paper conducts qualitative evaluation on some QA samples. Is there any generation task in financial domain that you can use to systematically evaluate FinPythia-6.9B?
2. This is mainly an empirical paper and does not have solid theoretical supports.
3. ETS gives better result than ETA, but I'm wondering if knowing task data is a reasonable assumption in the real world as a foundation model is mainly designed for multiple tasks.
4. The paper involves a lot of abbreviations that are quite similar, which makes the paper hard to read. I suggest using the full word "task-specific" and "task-agnostic" and removing the same suffix "DACP" in the abbreviations.

**Questions:**

1. The paper argues that good data selection can make continual pre-training data-efficient and maintain the performance on general tasks. Speaking of this part, I think [1] needs to be discussed in the paper. That work also studies the continual pre-training problem and considers preserving the model's general ability from a different perspective.
2. I think "building domain-specific LLMs from scratch" in the abstract may confuse the readers as continual pre-training is actually opposed to pre-training from scratch.

[1] Adapting a Language Model While Preserving its General Knowledge, Ke et al., EMNLP 2022

---

> ### Author Response · Authors · 2023-11-18
> **Discussion about Ke et al**
>
> We would first like to thank the reviewer for their insightful questions by reading the paper carefully and raising valid points.
>
> We would first like to respond to the weakness mentioned by the reviewer:
>
> * The NER task is a generation task. We did experiment on ConvFinQA task, which is a QA type task on Finance domain. However, all the methods scored near zero with it as this task needs a really long context window for generation. Most of the other tasks for Finance are similar to ones we experimented with.
> * We agree with the reviewer that the paper is mostly an empirical paper, though we did discuss the intuition behind our data selection in Section 2.4 with existing theory, which is indeed demonstrated in the empirical conclusions; similarity based data selection is indeed the best. This kind of work is first in the LLM domain where theory is limited.
> * We agree with the reviewer that knowing task data is not usually possible in real world. Hence, our proposed task-agnostic methods are much more useful and practical in the real world LLMs. We have added this discussion in Section 4.3 under *Comparison of Data Selection Metrics*. If there is a task or group of tasks available, similarity based method can be used.
>
> Response to Questions:
>
> * Thanks for pointing it out. We have added the discussion about Ke et al’s work in Section 5 under continual pre-training of language models section. In summary, Ke at al’s DGA [1] and DAS [2] works need multiple passes over the entire pre-training corpus, costing 3x of vanilla pre-training, which is infeasible for LLMs and the large amount of data used in our experiments.  Our methods pre filter the samples even before pre-training starts, while DGA/DAS filter the sample’s importance during the pre-training step. Our methods only cost 10% of vanilla pre-training: 3.33% of cost of training with DGA/DAS, which is quite substantial given the cost of training LLMs.
> * Thank you for pointing it out, we have reworded the abstract to make it more clear. Hope it is not confusing anymore to the readers.
>
>
> [1] Ke, et al. "Continual Pre-training of Language Models." ICLR. 2023.
>
> [2] Ke, et al. "Adapting a Language Model While Preserving its General Knowledge." EMNLP. 2022.

---

> > ### Comment · Reviewer_LVHJ · 2023-11-19
> >
> > I thank the authors for clarifying the first point I mentioned in "Weaknesses" and updating the manuscript in respond to my questions. I think this is very helpful for readers to understand this work better. However, I resonate with Reviewer QRSW that "the results of the proposed efficient-DACP methods are generally positive, but it is unclear to what extent those positive results might generalize to other domain-specific datasets" and the authors also agree that this is more of an empirical paper experimenting on a specific domain. Thus, I think the main contribution of this work would be the curated continual pre-training dataset and the trained model for the financial domain.
> >
> > Considering all this, I prefer to remain my previous rating unchanged as it indicates that I'm inclined to accept this work if room permitting.

---

> > > ### Author Response · Authors · 2023-11-21
> > >
> > > Thank you for your valuable feedback in making this work better, and inclination to accept our work. We really appreciate your time.

---

### Official Review · Reviewer_QRSW · 2023-10-30

**Soundness:** 2 fair
**Presentation:** 2 fair
**Contribution:** 2 fair
**Rating:** 5
**Confidence:** 3

**Summary:**

This paper explores domain-specific continual pretraining for LLMs through continued pretraining of Pythia models on a newly curated financial dataset. The authors demonstrate that domain-adaptive continual pretraininig (DACP) can improve performance of an LLM on domain-specific tasks. They then explore methods for improving the efficiency of DACP by selecting data according to different metrics, demonstrating efficiency and performance gains using only 10% of the tokens of the original domain-specfic dataset. Finally, the authors show that general (non-domain-specific) performance is degraded less in the 10% trained models compared to the 100% DACP baseline, suggesting that in addition to gains in domain-specific performance and in training efficiency, the proposed methods help the model retain more of its original domain-nonspecific performance.

**Strengths:**

The paper has an appealing overall structure (DACP vs baseline, modifications of DACP vs DACP on domain-performance, mod vs DACP on general performance). The questions addressed are of immediate topical relevance to researchers and practitioners of LLMs. The fact that the proposed modifications to DACP yield better specific performance and better generalization performance is a nice contribution.

**Weaknesses:**

The fundamental questions of the paper are valuable to address, but the conclusions drawn from the answers provided by this paper are not as comprehensive or original as one might hope. The demonstration that training a pretrained LLM with continued pretraining on a domain-specific dataset is more efficient than training from scratch is not a surprise, nor a novelty. The results of the proposed efficient-DACP methods are generally positive (though not universally, and the deficiencies are not explained or addressed), but it is unclear to what extent those positive results might generalize to other domain-specific datasets.

The clarity of the writing could be improved. Some sections (introduction of TACP) are ambiguous regarding important details, or just difficult to read (SoftSampling).

The results of the paper are only presented on 10% and 100% of the domains-specific corpus. Why 10% in particular? It would be useful to understand the tradeoffs involved at differing percentages. The paper would be strengthened by showing the performance curves across other percentages as well. Particularly, how does this perform with as little as 1%? At what dataset size does high-quality data fail improve domain specific performance in continual pretraining? This would be very interesting to know.

The results tables (3 and 4) are very busy and somewhat difficult to parse. The stddevs presented in Table 3 are really large, which makes it difficult to be confident in the significance of the results. The relationship between the columns of Table 1 and the rows of Table 3 is unclear. Overall, the results in the tables support the arguments made in the text, but the tables are difficult to understand due to their arrangement and formatting.

Different metrics are proposed for scoring the quality of the domain-specific data, and their performance varies dramatically across datasets. The paper would be made much stronger with analysis of the strengths and weaknesses of the different metrics, and by explanations of their divergent performance, perhaps with examples. As is, it is difficult to asses which metric one should use, and why. For example, ETA-DACP-ppl uses perplexity as a heuristic for novelty, which is assumed to be desirable. In a high-quality dataset, this might be a reasonable assumption, but in a noisy dataset this will likely score highly noisy examples, leading to a noise-enriched low quality sample. This method performs quite poorly compared to the others, suggesting the dataset may be noisy. To what extent are the other methods just avoiding training on the noisy examples in the full dataset?

**Questions:**

Questions/suggestions:

Why is performance on FiQA SA so bad? Table 1. This is not necessarily an issue but is an outlier compared to the other datasets so should be mentioned/explained.

TACP: Make explicitly clear the similarities/differences between TACP / DACP / generic finetuning while introducing TACP. What is the difference between domain-specificity and task-specificity? Is domain-specific LM data on a topic, and Task-specific is data formatted in a specific format (ie QA style)? From the text, it is unclear how DACP and TACP differ except in the amount of {domain/task}-specific data available, and this distinction isnt made until section 2.4, but should be made in 2.3. Also, is there a citation for TACP?

Soft Sampling (2.4.3): please rewrite this section to make more clear the methodology. The first couple sentences leave ambiguous whether the softness refers to you are probabilistic sampling or uniform sampling with example-loss-weighting, and the sentences that clarify that ambiguity aren't presented til midway through the next paragraph.

Table 1:
Put the columns for the 1B models and 7B models adjacent to each other, so the comparison is easily made. Consider also adding diffs for the FinPythia performance showing the delta compared to Pythia, as the diff is the point of the table.

Table 3:
The stddev of all models, even on the averaged F1 is very high. This makes it hard to know how significant the results are.
Shouldn't the Pythia 1B column in Table 1 correspond to the same evaluation as the Pythia 1B row in Table 3? Please clarify.
Which row does FinPythia 1B from Table 1 correspond to in Table 3? DACP 100%?

Table 4:
HellaSwag isn't mentioned anywhere else in the text of the paper. Please introduce all datasets in use here (probably in 3.1, optionally in 4.2). Also, why not present the Pile test loss as well?
Since the point of the table is to demonstrate low deltas over the baseline Pythia model, please show the actual deltas in the text of the table. This would help the table make its point and significantly improve readability.

Table 2: this takes up a ton of room and doesn't add much. Consider trimming to 2 examples or moving to appendix.

**Details Of Ethics Concerns:**

No ethics concerns.

---

> ### Author Response · Authors · 2023-11-18
> **Thanks for the great review. Rewritten and added new experiments.**
>
> We would like to thank the reviewer for such a thorough review with a careful reading of our work. We really appreciate your insights and have tried to address your valid concerns in the new revision. We hope these have improved the quality of our work significantly.
>
> **Response to the weakness by reviewer**
> * We have rewritten and added details in sections (listed below) as rightly pointed out by the reviewer. We have simplified the Tables 2, 3, and 4 based on the suggestions from the reviewer.
> * The results of the paper are only presented on 10% and 100% of the domains-specific corpus: We have added Figure 2, with ablation on % of pre-training data selected vs performance and discussion of these results (Section 4.3). In summary, we see a saturation of performance between 5%-10% of training data selection, which starts to degrade for both similarity based (ETS-DACP) and entropy based (ETA-DACP-ent) beyond 10%.
> * We agree with the reviewer on the high std deviation in Table 3. We actually found that standard deviation of results in Table 2 was higher as 5-shots are selected from the entire test set randomly and selected examples can make a huge difference on quality of results. To keep these numbers comparable to ones in the literature, we kept the evaluation framework same in Table 2. In order to make a fair comparison for data selection methods, we fixed the pool  of shot samples to 50, which is same across all the methods for our results in Table 3.  Hence, the difference in results for Pythia/FinPythia in Table 2 and 3. We have added a note on this in Table 3. This provided lower standard deviation, which is not typically reported by other works in the literature. However, we are confident on these conclusions because of trends (Table 3) and loss functions (Section C in Appendix) matching our intuitions, exception being perplexity based results, which are explained next with our investigation.
> * We have added a discussion of comparison between the different metrics in Section 4.3 under *Comparison of Data Selection Metrics*. Further, we have added Figure 3 in Appendix C that pictorially depicts task aware vs task agnostic data selection and tied it to an expanded discussion in Appendix Section C about loss curves for different metrics to show that loss curves have an expected behavior based on the way sampling is being done. In summary, to answer review’s question about  why perplexity based selection leads to poor performance , we observed a lot of long form tables in the top perplexity samples beyond the top 1% perplexity samples, which indeed can be noisy for the model to learn on, as pointed out correctly by the reviewer.  These samples drop the performance of perplexity based method (presented in ablation in Figure 2) until 5% of data selection, after which it starts to recover. In a higher quality dataset, perplexity could still be a good measure but owing to noisy samples scoring higher on perplexity/novelty, it is not a good metric for data selection. Your intuition is right that similarity and entropy based methods avoid these noisy examples as shown by the low correlation 0.21 and 0.14 with perplexity, respectively, in Figure 6 in Appendix.
>
>
> **Response to the questions**
> We have rewritten and added sections which were not clear as rightly pointed out by the reviewer:
>
> * FiQA SA Results: We believe that it is not an outlier. As we target a portfolio of tasks, the continual pre-training method may affect the tasks unevenly. Moreover, the mixing of the task dataset might play an uneven role in data selection as well.
> * TACP: We have expanded on this section and elaborated on the difference between TACP and DACP/fine-tuning. Further, we have added Table 1, which gives a summary of data selection metrics and data sources used by all the methods used in this work + Fine tuning.
> * Soft Sampling:  We have revised section (2.4.3) by explaining soft sampling as well as adding an example. We have also added a discussion about effect of soft versus hard sampling on the loss curves of our proposed methods in the Appendix C, to give a further intuition of how soft sampling makes a difference to what model sees - effect of which is quite prominent on previously confined ETS-DACP whose loss resembles other methods with soft sampling as it now sees the wider corpus distribution.
> * Simplified Table 2 (former Table 1), Table 3, and Table 4. We further added deltas in all the tables to make it easier for the readers to follow. Thanks for this suggestion, it really helps the reader without getting lost in individual columns.
> * We have added description about the four open domain tasks in Section 3.1 with relevant papers.
> * Former Table 2 with qualitative examples, has been moved to Appendix. Thanks for this suggestion, this freed up a lot of space to explain the unclear sections.

---

### Author Response · Authors · 2023-11-18
**General Response to the reviewers: Thanks for the insightful comments, we have made major changes**

We would really like to express our gratitude to all the reviewers for their insightful reviews. We have made significant changes to the paper, to better respond to your insightful comments. Major changes and clarifications are as:

* **Ablation experiment**: We have added ablation experiments in Figure 2, with percentage of pre-training data selected versus average F1 score for all the data selection methods with a discussion in Section 4.3. To summarize, we see a saturation of performance when 5%-10% of training data is selected, which starts to degrade for similarity based (ETS-DACP) and entropy based (ETA-DACP-ent) beyond 10%.
* **Comparison with Ke et al’s work**: We have added a discussion of comparison with Ke et al’s work [1, 2] in Section 5 (Related Work):

           1) Our method pre-filters the data before pre-training step while these methods calculate the sample importance during pre-training. Hence, they need to pre-train on all the samples.

            2) These works costs 300% of vanilla pre-training, while our method costs 10% of vanilla pre-training; our methods cost 3.33% or lower than these methods.

* **Novelty**: While there is a general belief in the LLM community that throwing more data at LLM would increase its performance. Our results show that the quality of selected data matters more than adding more data, especially for domain LLMs. While literature has studied effects of deduplication of pre-training on LLM performance, we present a novel insight about the quality of pre-training data. While our method is simple, it is more practical and cost effective for creating domain LLMs, which benefits both industry and academia with sensitive/private data for creating LLMs with their limited budgets.
* We have rewritten sections and re-arranged tables as pointed out by reviewers for better understanding. Please see comments to your reviews on the changes made.

[1] Ke, et al. "Continual Pre-training of Language Models." ICLR. 2023.

[2] Ke, et al. "Adapting a Language Model While Preserving its General Knowledge." EMNLP. 2022.